# Complete and efficient conversion of plant cell wall hemicellulose into high-value bioproducts by engineered yeast

Liang Sun [1,2,3], Jae Won Lee [1,2,3], Sangdo Yook[1,2,3], Stephan Lane[1,2,3], Ziqiao Sun[1], Soo Rin Kim[4] & Yong-Su Jin [1,2,3✉]

Plant cell wall hydrolysates contain not only sugars but also substantial amounts of acetate, a fermentation inhibitor that hinders bioconversion of lignocellulose. Despite the toxic and non-consumable nature of acetate during glucose metabolism, we demonstrate that acetate can be rapidly co-consumed with xylose by engineered *Saccharomyces cerevisiae*. The co-consumption leads to a metabolic re-configuration that boosts the synthesis of acetyl-CoA derived bioproducts, including triacetic acid lactone (TAL) and vitamin A, in engineered strains. Notably, by co-feeding xylose and acetate, an enginered strain produces 23.91 g/L TAL with a productivity of 0.29 g/L/h in bioreactor fermentation. This strain also completely converts a hemicellulose hydrolysate of switchgrass into 3.55 g/L TAL. These findings establish a versatile strategy that not only transforms an inhibitor into a valuable substrate but also expands the capacity of acetyl-CoA supply in *S. cerevisiae* for efficient bioconversion of cellulosic biomass.

[1] Department of Food Science and Human Nutrition, University of Illinois at Urbana-Champaign, Urbana, IL, USA. [2] Carl R. Woese Institute for Genomic Biology, University of Illinois at Urbana-Champaign, Urbana, IL, USA. [3] DOE Center for Advanced Bioenergy and Bioproducts Innovation, University of Illinois at Urbana-Champaign, Urbana, IL, USA. [4] School of Food Science and Biotechnology, Kyungpook National University, Daegu, South Korea. ✉email: ysjin@illinois.edu

The increasing demands for natural resources and rising concerns on climate change have ignited a broad interest to produce biofuels and chemicals via microbial fermentation using renewable biomass[1,2]. Second-generation bioenergy and biorefineries based on lignocellulosic plant materials could serve as a sustainable alternative to conventional petrochemical processes[3,4]. Under this goal, microbial engineers have exerted tremendous efforts to increase bioconversion efficiencies of cellulosic carbons into target bioproducts[5–8]. However, due to the acetylation of hemicellulose and lignin in the plant cell wall, cellulosic hydrolysates inevitably contain substantial amounts of acetate, an inhibitor toxic to fermenting microorganisms that negatively influences the bioconversion efficiency of cellulosic hydrolysates[9]. The toxicity of acetate against fermenting microorganisms is one of the major obstacles hampering the commercialization of lignocellulose-based biorefinery.

Although the generation of acetate can be reduced by genetic perturbations to modify plant cell wall[10], or prudent design of depolymerization processes of cellulosic biomass, complete elimination of acetate in cellulosic hydrolysates is unattainable. Alternatively, a wide variety of approaches have been proposed for physical, chemical, and biological removal of acetate before fermentation[11,12], but they impose additional costs to implement. In addition to these pre-fermentation strategies, efforts have been devoted to engineering acetate tolerance in yeast, alleviating but being unable to eliminate the inhibitory and toxic effects of acetate[13,14]. Previously, we introduced an NADH-dependent acetate reduction pathway into engineered yeast for in situ detoxification of acetate and improved ethanol production from a mixture of xylose and acetate under anaerobic conditions[7]. However, the capacity of the acetate reduction pathway might be limited by the activities of key enzymes, ATP supply, and intracellular NADH levels[15]. Besides, the coupled acetate reduction and xylose assimilation also limit its application to cellulosic ethanol production.

Enabling efficient acetate detoxification and valorization by *S. cerevisiae* would greatly facilitate industrial cellulosic biorefineries due to the versatility and robustness of *S. cerevisiae* as an industrial production platform. Conversion of acetate into acetyl-CoA through a one-step enzymatic reaction, and production of high-value acetyl-CoA-derived molecules, such as fatty acids, sterols, polyketides, and isoprenoids have been demonstrated in oleaginous yeasts[16–19]. While the model yeast *S. cerevisiae* can produce acetyl-CoA from acetate[20], there has been no report of producing acetyl-CoA derivatives from acetate by *S. cerevisiae* due to toxicity of acetate and slow acetate metabolism. Dedication of *S. cerevisiae* to fermentative production of ethanol from glucose overshadowed the fact that it can grow on acetate as a sole carbon source under aerobic and neutral pH conditions[21]. In such cases, acetate is taken up by yeast cells through either Jen1 transporter[22] or Ady2 symporter[23], and converted into acetyl-CoA by acetyl-CoA synthase (Acs1 and Acs2). Acetyl-CoA is then metabolized through the tricarboxylic acid (TCA) cycle or glyoxylate shunt for the generation of ATP and precursors toward gluconeogenesis[24]. Nonetheless, the respiratory metabolism of acetate as a sole carbon source may lead to unfavorable growth kinetics due to the generation of harmful reactive oxygen species (ROS)[25] and unbalanced ATP supply-and-demand[26]. Additionally, acetate is toxic to yeast cells at low pH (<pKa = 4.76) conditions due to the ATP-dependent translocation of protons generated by intracellular dissociation of acetic acid[27,28]. Most importantly, glucose inhibits the transport and metabolism of acetate regardless of aerations so that acetate is trapped as a toxic inhibitor of glucose fermentation[21].

Here, we focus on detoxifying acetate in cellulosic hydrolysates and exploiting its consumption to enhance acetyl-CoA supply in

*S. cerevisiae* for the production of value-added products. We demonstrate that xylose, unlike glucose, does not impose repression on acetate transport and metabolism, which enables efficient co-consumption of xylose and acetate under aerobic conditions. Additionally, the co-consumption substantially improves cell growth and accumulations of lipids and ergosterol, suggesting enhanced supply of acetyl-CoA when xylose and acetate are co-consumed. We, therefore, envision aerobic co-utilization of xylose and acetate, which are major carbon sources in hemicellulose hydrolysates, as a potential strategy for economically producing acetyl-CoA derivatives. In this simple metabolic design, xylose metabolism generates NADPH and ATP to support cell growth and acetate assimilation while acetate is directed toward acetyl-CoA production (Fig. 1). Such a division of duty may also ward off the negative effects of respiratory acetate metabolism while acetate is used as a sole carbon source. As a proof of concept, we choose a polyketide triacetic acid lactone (TAL) and an isoprenoid vitamin A as examples of high-value acetyl-CoA derived chemicals and achieve significantly improved titers of both products through co-consumption of xylose and acetate. Thus, we establish a strategy that detoxifies acetate into a valuable substrate, expands the capacity of acetyl-CoA supply in *S. cerevisiae*, and enables economic conversion of plant cell wall hydrolysates into acetyl-CoA derived fine chemicals.

## Results

**Xylose enables efficient aerobic co-consumption of acetate in *S. cerevisiae*.** Previously, we constructed an engineered yeast strain SR8 capable of assimilating xylose efficiently through rational engineering and adaptive laboratory evolution[29]. Based on this strain, we investigated interactions of glucose or xylose with acetate assimilation under aerobic conditions by adding acetate into glucose or xylose cultures at different concentrations. Expectedly, glucose metabolism exhibited strong inhibition on acetate consumption so that acetate was not consumed in the cultures with 40 g/L of glucose (Fig. 2a). Instead, every tested acetate concentration negatively affected cell growth and ethanol re-assimilation in the glucose cultures (Fig. 2a, c). The inhibited ethanol re-assimilation was partially attributed to the disruption of *ALD6* gene involved in ethanol utilization as the parental strain SR7 with wildtype *ALD6* respired ethanol much faster than the SR8 strain (Supplementary Fig. 1). In contrast to the hampered acetate consumption by glucose, up to 15 g/L of acetate was assimilated by engineered yeast in the xylose cultures (Fig. 2b). Notably, xylose consumption was rather slightly boosted than hindered by the addition of up to 12.38 g/L of acetate, allowing efficient co-consumption of xylose and acetate. These results reveal that xylose metabolism empowers efficient co-consumption of acetate in *S. cerevisiae* under aerobic conditions.

**Factors facilitating xylose and acetate co-consumption.** To determine the mechanisms enabling xylose and acetate co-consumption by engineered yeast, we compared transcriptional levels of the genes (*ACS1*, *ACS2*, *ADY2*, *JEN1*, *FPS1*, *PMA1*, and *HOG1*) related to acetate assimilation under glucose and xylose conditions. The expression levels of *ADY2* and *JEN1*, coding for monocarboxylic acid transporters, were significantly higher under xylose than glucose conditions (Fig. 3), indicating that xylose alleviates glucose repression on acetate transport[21]. Moreover, *ACS1* coding for a major isoform of acetyl-CoA synthase was highly expressed when cells were grown on xylose as compared to glucose (Fig. 3), facilitating the production of acetyl-CoA from acetate under xylose conditions. Nevertheless, the mRNA level of *PMA1*, coding for a proton-translocating ATPase in the plasma

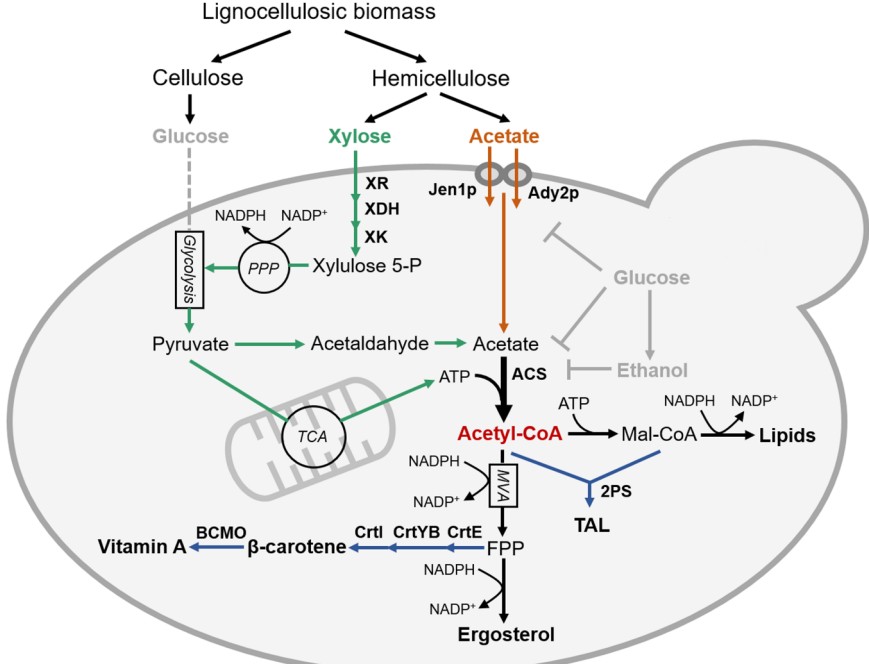

**Fig. 1 Schematic diagram of the production of TAL and (pro-)vitamin A from hemicellulose fractions by engineered *S. cerevisiae*.** Xylose is assimilated through the oxidoreductive pathway consisting of XR (xylose reductase), XDH (xylitol dehydrogenase), and XK (xylulose kinase). Acetate enters yeast cells through either Jen1p or Ady2p transporter and is converted into acetyl-CoA by ACS (acetyl-CoA synthase). TAL is synthesized through the condensation of acetyl-CoA and malonyl-CoA by 2PS (2-pyrone synthase). Heterologous expression of CrtE (GGPP synthase), CrtYB (bifunctional phytoene synthase, lycopene cyclase), and CrtI (phytoene desaturase) in *S. cerevisiae* allows the production of β-carotene which is then cleaved into vitamin A by BCMO (β-carotene monooxygenase). Color code for arrows: gray, glucose metabolism; green, xylose metabolism; orange, acetate metabolism; black, other endogenous pathways; black in bold, reaction with strong flux; blue, heterologous TAL or vitamin A biosynthetic pathway.

membrane, was substantially lower on xylose than glucose (Fig. 3).

When glucose and acetate were provided, yeast cells rapidly convert glucose into ethanol due to the Crabtree effect[30]. However, they were unable to subsequently consume acetate (Fig. 2). We, therefore, examined how ethanol affects acetate assimilation, and found that acetate consumption was impeded in the presence of ethanol at a concentration higher than 4.25 g/L (Supplementary Fig. 2). The ethanol-dependent repression was not observed in xylose cultures due to inactivated overflow metabolism where the overflow of glycolytic flux into ethanol production does not occur[31] (Fig. 2b). Additionally, despite buffering culture media at an initial pH of 5.50, consumption of glucose lowered pH of culture media to the levels (<pH 4.6) at which acetate becomes toxic to yeast cells (Supplementary Fig. 3). In contrast, initial acetate concentrations are positively correlated with final media pH when yeast co-utilizes xylose and acetate (Supplementary Fig. 3). Altogether, these results indicate that the combined effects of inactivated overflow metabolism and bypassed glucose repression of acetate assimilation under xylose conditions might facilitate co-consumption of xylose and acetate. The consumption of acetate increases media pH and benefits xylose assimilation in return[7].

**A potential strategy to expand the supply of acetyl-CoA in *S. cerevisiae*.** As a result of xylose and acetate co-consumption by engineered yeast, we observed significant improvements in cell concentrations in the xylose cultures with acetate supplementation up to 12 g/L (Fig. 2c). Xylose and acetate co-consumption also induced substantial increases in both lipid and ergosterol contents in proportion to the supplemented acetate (Fig. 2c). Notably, the cells from the cultures that completely consumed 40 g/L of xylose and 12 g/L of

acetate accumulated lipids at a specific content of 71.93 ± 2.07 mg/g dry cell weight (DCW), roughly 48% higher than the cultures with 40 g/L of xylose (48.64 ± 4.70 mg/g DCW) only. Additionally, the cells accumulated 26.90 ± 1.31 mg/g DCW of ergosterol when co-consuming 12 g/L of acetate with 40 g/L of xylose, roughly 45% higher than the culture grown with xylose as a sole substrate. Considering that lipids and ergosterol are derived from a common precursor, acetyl-CoA, these results indicate that co-utilization of xylose and acetate facilitates an ample supply of acetyl-CoA in *S. cerevisiae*, which also contributes to enhanced accumulation of cell biomass.

**Establishing and improving TAL production in *S. cerevisiae*.** We next aimed to harness this enhanced acetyl-CoA supply during xylose and acetate co-consumption for the production of TAL, a versatile platform chemical for producing various polymers and food ingredients[19]. Specifically, the codon-optimized *Gerbera* 2-pyrone synthase gene, *2PS*, was expressed in a xylose-fermenting *S. cerevisiae*, resulting in the Tal1 strain capable of producing TAL from glucose and xylose. As expected, the Tal1 strain produced TAL from xylose at a titer (198.46 mg/L) about 6-fold higher than from glucose (33.24 mg/L) (Fig. 4a). By increasing the copy number of the *2PS* gene expression cassettes up to four copies, we obtained the Tal4 strain exhibiting enhanced TAL production from both glucose (539.87 mg/L) and xylose (1522.44 mg/L) (Fig. 4a, and Supplementary Figs. 4, 5).

The Tal4 strain was cultured in different single- and mixed-carbon source conditions to investigate their carbon flux distributions toward acetyl-CoA. In terms of single-carbon source conditions, acetate exhibited the highest yield of TAL at 107.81 mg/g acetate, followed by xylose (47.47 mg/g xylose), and ethanol (40.51 mg/g ethanol) (Fig. 4b, and Supplementary Figs. 4, 5). When cultured on 40 g/L of xylose with supplementation of 10 g/L

**a**

**b**

**c**

**Fig. 2 Co-utilization of xylose and acetate by the xylose-fermenting SR8 strain under aerobic conditions. a** Glucose cultures with acetate supplementation of 0, 6, 12, and 16 g/L. **b** Xylose cultures with acetate supplementation of 0, 6, 12, and 15 g/L. **c** Improved accumulation of cell biomass, lipid, and ergosterol as a result of xylose and acetate co-utilization (Cell biomass: ***$P < 0.001$ 5 vs 10 g/L acetate with glucose, ***$P < 0.001$ 10 vs 5 g/L acetate with glucose, **$P = 0.005$ 5 vs 0 g/L acetate with xylose, ***$P = 0.001$ 10 vs 0 g/L acetate with xylose; Lipid content: ***$P = 0.001$ xylose vs glucose culture, ***$P < 0.001$ 5 vs 0 g/L acetate with xylose, *$P = 0.047$ 10 vs 5 g/L acetate with xylose; Ergosterol content: *$P = 0.024$ 5 vs 0 g/L acetate with xylose, ***$P = 0.003$ 10 vs 0 g/L acetate with xylose). Data are presented as mean value and standard deviations of three independent biological replicates. Statistical significance of the differences was evaluated by one-way ANOVA followed by Tukey's multiple-comparison test. Source data are provided as a Source Data file.

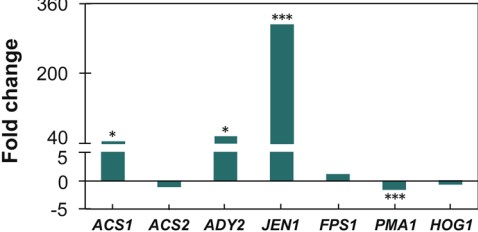

**Fig. 3 Transcriptional patterns of genes related to acetate transport and metabolism of the xylose-fermenting SR7 strain on glucose and xylose.** Fold changes were calculated by dividing expression levels of genes on xylose by those on glucose ($n = 3$ biological independent samples, *$P_{ACS1} = 0.03$, *$P_{ADY2} = 0.04$, ***$P_{JEN1} = 0.0017$, ***$P_{PMA1} = 0.0011$). Statistical significance of the differences was evaluated using two-tailed Student's $t$-tests. Source data are provided as a Source Data file.

of acetate, the Tal4 strain produced 4936.77 mg/L of TAL at a yield of 90.24 mg/g, roughly a 3-fold and 2-fold improvement over cultures with 40 g/L of xylose as a sole substrate (Fig. 4b, c, and Supplementary Fig. 6c). Even though up to 20 g/L of acetate could be completely depleted with 40 g/L of xylose, the best titer and

productivity were obtained in cultures starting with 10 g/L of acetate provided (Fig. 4c, and Supplementary Fig. 6). Therefore, a 4:1 ratio of xylose and acetate concentrations, which is also a typical ratio of xylose and acetate concentrations in cellulosic hydrolysates, was adopted for the following fermentation experiments. These data demonstrate that xylose and acetate co-consumption is an effective strategy to overproduce TAL.

**TAL production in a bioreactor with xylose and acetate co-feeding.** To maximize the titer of TAL and explore the feasibility of the strategy in a large-scale fermentation, the Tal4 strain was cultured in a 3-L bioreactor with xylose and acetate co-feeding. The fed-batch fermentation produced 23.89 g/L of TAL with a volumetric productivity of 0.29 g/L/h (Fig. 5). As the production level far exceeded the solubility of TAL (8.6 g/L), we noticed precipitation of TAL in the culture during the fermentation and adopted cautious sampling and analysis to avoid errors in TAL measurements. Glycerol at 2.79 g/L remained the only byproduct at the end of fermentation (Fig. 5a), which may result in sim-plified purification steps. Noticeably, cell growth slowed down after 56 h when TAL concentration reached up to 9.51 g/L

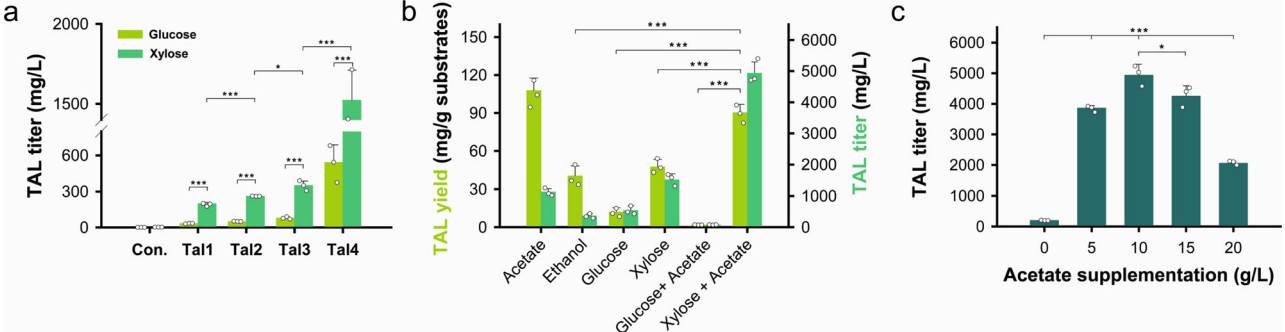

**Fig. 4 Effect of 2PS gene copy numbers and substrates on TAL production. a** Improved production of TAL by increasing the copy numbers of 2PS genes (***$P < 0.001$ Tal1 xylose vs glucose, ***$P < 0.001$ Tal2 xylose vs glucose, ***$P < 0.001$ Tal3 xylose vs glucose, ***$P = 0.002$ Tal4 xylose vs glucose, ***$P < 0.001$ Tal2 vs Tal1 in xylose, *$P = 0.014$ Tal3 vs Tal2 in xylose, ***$P < 0.001$ Tal4 vs Tal3 in xylose). **b** Yield of TAL from different single and mixed substrates by the Tal4 strain. Calculated from cultures of Tal4 strain with 10 g/L ethanol, 10 g/L acetate, 40 g/L glucose, 40 g/L xylose, 40 g/L glucose + 10 g/L acetate and 40 g/L xylose + 10 g/L acetate, respectively (TAL Yield: ***$P < 0.001$ Xylose + Acetate vs Glucose + Acetate, ***$P < 0.001$ Xylose + Acetate vs Glucose, ***$P < 0.001$ Xylose + Acetate vs Ethanol, ***$P < 0.001$ Xylose + Acetate vs xylose). **c** Production of TAL by the Tal4 strain from 40 g/L xylose with supplementation at varying amounts of acetate (***$P < 0.001$ 5 vs 0 g/L, ***$P = 0.001$ 10 vs 5 g/L, *$P = 0.025$ 10 vs 15 g/L, ***$P < 0.001$ 10 vs 20 g/L). Data are presented as mean value and standard deviations of three independent biological replicates. Statistical significance of the differences was evaluated by one-way ANOVA followed by Tukey's multiple-comparison test. Source data are provided as a Source Data file.

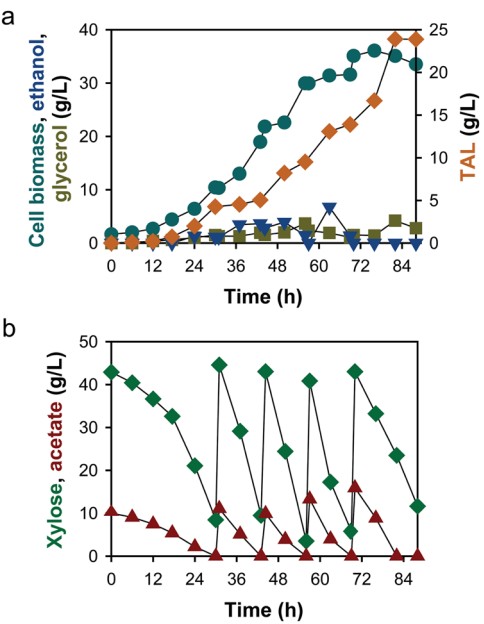

**Fig. 5 Fed-batch culture of the Tal4 strain in a 3-L bioreactor with xylose and acetate co-feeding. a** Profiles of TAL, cell biomass, ethanol, and glycerol concentrations. **b** Feeding patterns of xylose and acetate. Source data are provided as a Source Data file.

(Fig. 5a). This may be caused by the toxicity of TAL to yeast cells at high concentrations[32].

**Conversion of hemicellulose fractions of switchgrass into TAL.** As xylose and acetate are predominant components of hemicellulose hydrolysates which can be readily obtained from lignocellulose materials, we next attempted bioconversion of hemicellulose fractions of switchgrass into TAL. Without any detoxification steps, the obtained switchgrass hemicellulose hydrolysates were used to cultivate the Tal4 strain. When cultured in two times concentrated hydrolysate, the Tal4 strain completely consumed xylose, acetate, and glucose within 115 h, and produced 3545.74 mg/L of TAL with a yield of 67.45 mg/g substrates (Fig. 6). Notably, 4.92 g/L glycerol was accumulated by

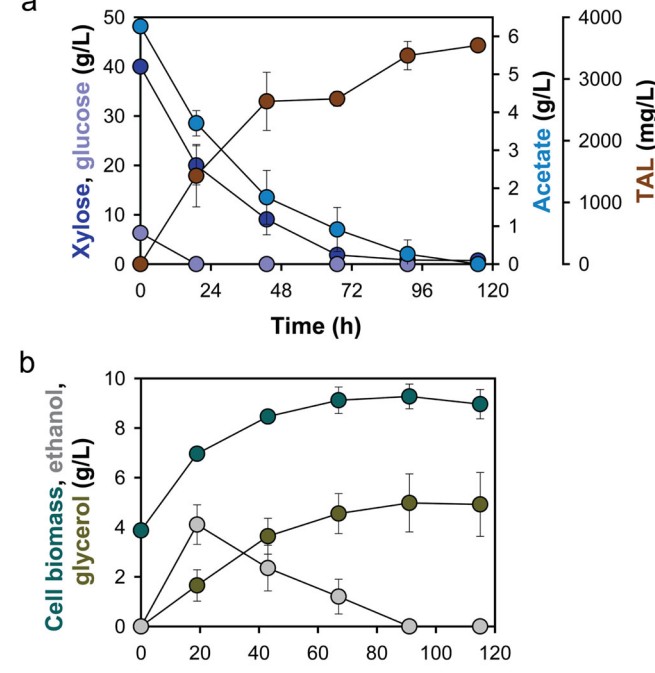

**Fig. 6 Production of TAL by the Tal4 strain in hemicellulose fractions of switchgrass hydrolysate. a** Profiles of TAL, xylose, acetate, and glucose concentrations. **b** Profiles of cell biomass, ethanol, and glycerol concentrations. Data are presented as mean value and standard deviations of three independent biological replicates. Source data are provided as a Source Data file.

the end of fermentation, possibly due to osmotic stress caused by the concentration and neutralization procedures[33]. When cultured in unconcentrated hydrolysate, the strain did not accumulate glycerol at the end of fermentation, leading to an overall higher yield of TAL (86.53 mg/g substrates) (Supplementary Fig. 7). We did not observe xylitol accumulation in this and all other xylose culturing experiments, suggesting sufficient oxygen supply to eliminate xylitol production. These results demonstrate

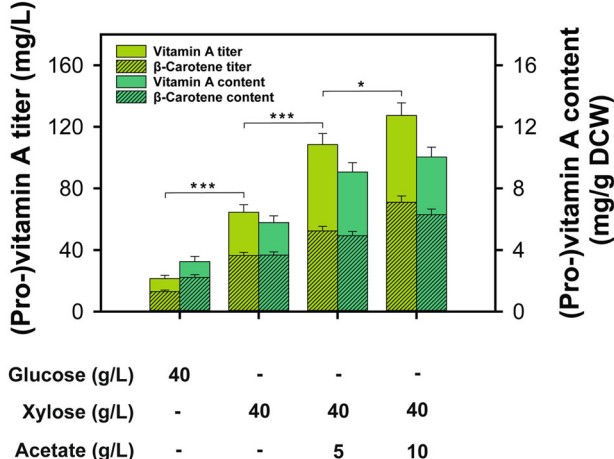

**Fig. 7 Improved production of (pro-)vitamin A by the SR8A strain via xylose and acetate co-utilization.** \*\*\*P < 0.001 xylose vs glucose, \*\*\*P < 0.001 5 vs 0 g/L acetate with xylose, \*P = 0.021 15 vs 10 g/L acetate with xylose. Data are presented as mean value and standard deviations of three independent biological replicates. Statistical significance of the differences was evaluated by one-way ANOVA followed by Tukey's multiple-comparison test. Source data are provided as a Source Data file.

the feasibility of complete conversion of hemicellulose fractions into acetyl-CoA derived chemicals using engineered *S. cerevisiae*.

**Enhanced vitamin A production via co-utilization of xylose and acetate.** After demonstrating efficient TAL production from hemicellulose fractions, we investigated the effectiveness of this strategy for the production of vitamin A, another valuable acetyl-CoA derivative. Vitamin A is an essential human micronutrient that has been widely used in dietary supplements, animal feed, cosmetics, and pharmaceuticals. A previously constructed xylose-fermenting, vitamin A producing strain SR8A[34] was used, and cultured on glucose or xylose with supplementation of acetate at varied concentrations. The co-consumption of 5 g/L and 10 g/L of acetate with 40 g/L of xylose resulted in 68.38 % and 97.83 % improvement in (pro-)vitamin A (vitamin A and β-carotene) titers as compared to the culture using xylose as a sole substrate (64.38 mg/L) (Fig. 7, and Supplementary Fig. 8). The best titer obtained here (127.37 mg/L) was 38.90 % higher than the previous report of (pro-)vitamin A production in a xylose flask culture with dodecane in situ extraction[34]. The improvement was partially derived from the increased cell biomass titers. Nonetheless, the specific content of (pro-)vitamin A was significantly enhanced, indicating enhanced acetyl-CoA availability when acetate is co-consumed with xylose.

## Discussion

In this study, we present efficient xylose-acetate co-consumption and its implementation as a metabolic engineering strategy to expand acetyl-CoA supply in *S. cerevisiae* for the enhanced production of acetyl-CoA derived chemicals. The simplest polyketide TAL and an isoprenoid vitamin A were selected as testbed molecules to examine this strategy.

Acetate transport and metabolism are both subject to glucose repression even in the presence of oxygen[21]. Apart from glucose repression, high concentrations of ethanol derived from glucose fermentation were found to constrain acetate assimilation by engineered yeast (Fig. 2a, and Supplementary Figs. 1, 2), possibly resulting from repressed *ACS1* transcription on ethanol[35]. As such, when glucose is used by yeast, acetate in the cell culture will not be consumed and will become toxic as pH of

the culture drops with glucose consumption (Fig. 2a, c). In contrast to glucose metabolism, xylose metabolism in *S. cerevisiae* is believed to be Crabtree-negative, resulting in inactive overflow metabolism, and active respiration under aerobic conditions[31,36]. This unique trait of xylose metabolism by engineered *S. cerevisiae* opens the door for efficient xylose-acetate co-consumption due to the alleviation of both glucose and ethanol repression on acetate assimilation (Figs. 2b, 3). Accordingly, up to 12.38 g/L of acetate was co-assimilated with 40 g/L of xylose at a rate of 0.23 g/L/h (Fig. 2b). The capacity and rate of acetate consumption are roughly 7-fold and 11-fold higher, respectively than those of the anaerobic acetate reduction pathway coupled with xylose assimilation which reduced acetate into ethanol[7].

*S. cerevisiae* has been widely used for ethanol production for thousands of years, but this yeast is recognized to bear a limited supply of acetyl-CoA from glucose due to the Crabtree effect[31,37]. Employing xylose as a carbon source instead of glucose offers an approach to address this limiting factor for enhanced production of acetyl-CoA-derived chemicals[34,38,39]. Inspired by the enhanced accumulation of lipids and ergosterol in the cells grown on a mixture of xylose and acetate (Fig. 2c), we speculated that aerobic co-consumption of xylose and acetate might further synergistically improve acetyl-CoA supply. Specifically, xylose metabolism sustains cell growth and provides ATP to support the conversion of acetate to acetyl-CoA as a metabolic shortcut (Fig. 1). On the other hand, the consumption of acetate increases medium pH and slightly boosts xylose consumption in return (Supplementary Fig. 3). Such division of duty also wards off the negative effects of acetate respiration when used as a single substrate.

As a proof of concept, we established TAL production from xylose in an engineered *S. cerevisiae* strain and observed enhanced production by co-utilization of xylose and acetate in both shake flask and bioreactor cultures. This methodology led to an increase in TAL titer (23.89 g/L) and overall productivity (0.29 g/L/h) by greater than six-fold and fourteen-fold, respectively, compared to the titer and productivity reported previously using engineered *S. cerevisiae*[40]. Furthermore, the productivity achieved here is over two-fold higher than a previous report using an engineered *Yarrowia lipolytica*[19], which is believed to harbor high flux through acetyl-CoA[19]. We also employed this strategy for the production of vitamin A (Fig. 7), indicating its versatility and effectiveness for overproducing many other high-value acetyl-CoA derived bioproducts in *S. cerevisiae* such as n-butanol, (poly) hydroxybutyrate, fatty acids, flavonoids, and other polyketides and isoprenoids.

More importantly, this strategy can be seamlessly integrated into cellulosic biomass-based biorefineries as xylose and acetate are predominant components of hemicellulose hydrolysates; we found that hemicellulose fractions of plant cell wall hydrolysates can be completely converted into acetyl-CoA derived fine chemicals (Fig. 6). Instead of acting as an inhibitor, acetate becomes a valuable substrate when coupled with its hemicellulose counterpart under aerobic conditions. Following this concept, cellodextrins could also allow acetate co-consumption considering intracellular hydrolysis of cellodextrins into glucose might alleviate glucose repression on acetate transport[8,41], enabling a more attractive bioconversion of cellulosic biomass. Moreover, these results demonstrate the feasibility of coupling sugar metabolism with organic acid assimilation in *S. cerevisiae*, opening the door to many potential applications. For instance, acid whey, a byproduct from the yogurt and cheese industries that contains substantial amounts of lactose and lactic acid could be harnessed to produce value-added products in engineered yeast. Additionally, as the cytosolic accumulation of citric acid and an inactive overflow metabolism are required to shift the metabolism of *S. cerevisiae* from ethanol fermentation to lipogenesis[42], we envision that

coupling citrate assimilation with xylose metabolism may facilitate lipid overproduction in *S. cerevisiae*. Metabolic optimization using a mixture of substrates has been proposed as a unique strategy to enhance the performance of microbes. Together with current metabolic engineering approaches, the co-substrate strategy would circumvent many bottlenecks associated with the complex and highly regulated sugar metabolism facilitating enhancements in yield and productivity[43]. In the case of using sugar and organic acid as mixed-carbon sources, sugar acts as a primary substrate to support cell growth while organic acid is added as a second substrate that provides a shortcut towards key metabolic intermediate such as acetyl-CoA. With proper design of sugar-organic acid combinations and feeding patterns, the principal biosynthetic components, carbon, ATP, and reducing equivalents, could be better balanced for an optimal production level exceeding the theoretical maximum of a single substrate fermentation.

Collectively, this work demonstrates a versatile strategy that detoxifies a ubiquitous inhibitor in plant cell wall hydrolysates into a valuable substrate, overcoming the limit of native metabolism for the supply of acetyl-CoA in *S. cerevisiae*, and enabling economically sustainable conversion of cellulosic carbon into high-value acetyl-CoA derived bioproducts. Based on these results, we envision that co-utilization of sugars and organic acids can be a paradigm of bioconversion in addition to traditional sugar-only or organic acid-only utilization for the production of biofuels, chemicals, and bioproducts.

## Methods

**Plasmid and strain construction**. The strains, plasmids, and oligonucleotides used in this study are listed in Supplementary Table 1, 2, 3, respectively. Standard protocols were used for molecular biology procedures[44]. The amino acid sequence of *Gerbera hybrida* 2-pyrone synthase obtained from UniProtKB (accession number P48391.2) was codon-optimized and synthesized as a gBlock by IDT (Integrated DNA Technologies, USA). The acquired gBlock-co2PS was amplified using primers 2PS-Amp-U and 2PS-Amp-D and cloned into the *XhoI* and *BamHI* sites of the pRS426-pCCW12 plasmid, resulting in the plasmid pRS426-2PS. The *2PS* expression cassette flanked by a strong constitutive *CCW12* promoter and *CYC1* terminator was cloned into yeast integrative plasmids pRS403, pRS404, pRS405, and pRS406 using CloneEZ® PCR Cloning Kit (GenScript Biotech, USA), resulting in pRS403-2PS, pRS404-2PS, pRS405-2PS, and pRS406-2PS, respectively. The primer sets Insert-U/Insert-D and Vector-U/Vector-D were used for the PCR cloning.

To construct TAL producing strains, *TRP1* was disrupted by Cas9-based genetic modifications[45] as an extra auxotrophic marker in the CT2-auxo strain[46]. Donor DNA was amplified using primers TRP1donor-U and TRP1donor-D. The plasmid CAS9-NAT (Addgene#64329) was transformed into the CT2-auxo strain followed by the guide RNA plasmid gRNA-trp-HYB and donor DNA transformation. Cells were selected on yeast extract-peptone (YP) plate (10 g/L yeast extract, 20 g/L peptone) containing glucose (YPD) supplemented with 120 μg/mL nourseothricin and 300 μg/mL Hygromycin B (YPDNH). The positive colonies with *TRP1* deletion were confirmed by sequencing using primers TRP1-Seq-U and TRP1-Seq-D and designated as the CT2-4 strain. The integrative plasmids pRS403-2PS, pRS406-2PS, pRS405-2PS, and pRS404-2PS were linearized using restriction enzymes *NdeI*, *NcoI*, *AflII*, and *PmlI* (New England Biolabs, USA), respectively, and sequentially transformed into CT2-4 strain. Cells were selected on synthetic complete (SC) plate minus the appropriate auxotrophic compounds, yielding Tal1, Tal2, Tal3, and Tal4 strains with different copy numbers of the *2PS* gene. The empty plasmids pRS403, pRS404, pRS405, and pRS406 were used to construct a prototrophic strain CT2-con.

**Media and culture conditions**. For shake flask fermentation, yeast cells were routinely pre-cultured in YPD medium and inoculated into main fermentations. The initial cell optical density ($OD_{600nm}$) of fermentations for TAL production in defined medium or unconcentrated switchgrass hemicellulose hydrolysate was ~1. All other flask fermentations started with an initial $OD_{600nm}$ of ~10. Main fermentations were conducted with 50 mL of modified Verduyn medium[34,47] containing glucose, xylose, acetate (potassium acetate), or ethanol as a carbon source in 250 mL baffled flasks at 30 °C and 300 rpm. Appropriate amino acids were added while culturing auxotrophic strains. Fermentation media were buffered with 50 mM potassium hydrogen phthalate at pH of 5.50. Fed-batch fermentation for TAL production was performed by inoculating yeast cells grown on YPD into a 3-liter bioreactor (New Brunswick Scientific-Eppendorf, Enfield, CT) containing 1 L of Verduyn medium with ~ 40 g/L xylose and ~ 10 g/L acetate (potassium acetate).

The initial optical density of cells was 4.00. Additional xylose and acetate were added to the bioreactor up to initial concentrations each time upon depletion. The pH was maintained at 5.5 automatically by pumping in 4 M NaOH, or 4 M HCl. The temperature was held constant at 30 °C with agitation adjusted from 3 Lpm (lines per minute) to 5 Lpm to maintain aerobic conditions according to the cell densities. Antifoam 204 (Sigma-Aldrich, St. Louis, US) was added to prevent the formation of foam.

**Preparation and conversion of switchgrass hydrolysate**. The switchgrass was harvested from the University of Illinois Energy Farm and shredded into small pieces. Dilute acid hydrolysis was then performed at 10% solids loading in 2% sulfuric acid. The hydrolysis was conducted using a sterilization cycle of 121 °C for 60 min in a laboratory steam sterilizer (Amsco® Lab 250, Steris, Ireland). The liquid fraction was separated from solids by filtration and neutralized using calcium hydroxide to pH 5.5. The resulting hemicellulose hydrolysate was either concentrated two times before or directly used for the production of TAL by the Tal4 strain in a 250 mL baffled flask with supplementation of 2 g/L yeast extract and 2 g/L peptone as a nitrogen source. The two times concentrated hydrolysate contained 40.00 g/L of xylose, 6.26 g/L of acetate, 6.31 g/L of glucose, 2.01 g/L of furfural, and 0.42 g/L of hydroxymethylfurfural (HMF). The fermentation was conducted in 50 mL of the fertilized hydrolysate at 30 °C and 300 rpm with an initial cell optical density of ~1 for nonconcentrated hydrolysate or ~10 for two times concentrated hydrolysate.

**RNA sequencing**. The rationally engineered xylose-fermenting *S. cerevisiae* SR7 which is the parental strain of SR8 and has a clear genetic background was used for RNA sequencing experiments to rule out the potential effects of the evolved mutations in SR8[29]. The SR7 strain was cultured in 50 ml YP media containing either 40 g/L glucose or 40 g/L xylose at initial $OD_{600nm}$ of ~0.1. The cultures were incubated in 250 ml flasks at 30 °C and 100 rpm. Cells were sampled in triplicate when $OD_{600nm}$ reached 1 for RNA extraction. By the time of sampling, there was around 37 g/L glucose or 33 g/L xylose remained in the media (Supplementary Fig. 9). Total RNA was extracted using YeaStar RNA Kit (Zymo Research, USA), and the quality of the RNA samples was evaluated via gel electrophoresis. Transcription patterns of SR7 strain on glucose and xylose were determined through RNA sequencing which was conducted on an Illumina HiSeq 2000 system at the W. M. Keck Center for Comparative and Functional Genomics at the University of Illinois at Urbana-Champaign. QIAGEN CLC Genomics Workbench 6.5 was used to analyze the RNA sequencing results. Paired-end 100 bp reads were trimmed and mapped to the *S. cerevisiae* S288C reference genome. The total number of exonic reads per kilobase million (RPKM) from three biological replicates of xylose culture were compared to those of glucose culture. Statistical significance of the differences was evaluated using two-tailed Student's *t* tests.

**Quantitative analysis**. Cell growth was monitored by measuring $OD_{600}$ using a spectrophotometer (BioMate 5; Thermo Fisher Scientific, Waltham, USA). A conversion factor of 0.41 was adopted to determine DCW from optical density[34]. The concentrations of glucose, xylose, xylitol, acetate, glycerol, and ethanol in culturing media were detected by high-performance liquid chromatography (HPLC, Agilent 1200 Series, Agilent Technologies, Wilmington, US) equipped with a refractive index detector and the Rezex ROA-Organic Acid H + (8%) column (Phenomenex Inc, Torrance, CA). Culturing media were diluted and analyzed at 50 °C with 0.005 M $H_2SO_4$ as the mobile phase. The flow rate of the mobile phase was set at 0.6 mL/min. Data were collected by Agilent ChemStation B.03.

Total lipid weight was determined following a previously reported protocol[48]. Specifically, 2 mL cell cultures with cell optical density of 10 were centrifuged at 21,130 × *g* for 1 min. Cell pellets were transferred into 15-mL glass tubes and crushed using BeadBeater with 6 mL of chloroform/methanol (1:1 volumetric). The samples were then mixed with 1.5 mL water and vortexed for 1 min. After centrifugation, the organic layer was collected, washed with 1.5 mL of 0.1% (w/v) NaCl water solution, and dried overnight at room temperature in a preweighed tube. The tube was further dried in an oven at 80 °C until they reached a constant weight to determine lipid content. Total lipid weight was calculated from the final tube weight by subtraction of the original tube weight. Total lipid weight was divided by DCW to obtain specific lipid content.

For ergosterol quantification, cell pellets were saponified and extracted using n-heptane for HPLC measurement[49]. Briefly, cells harvested from 4 mL of fermentation broth were mixed with 0.6 mL extraction solution (50% KOH: $C_2H_5OH = 2:3$) and incubated at 85 °C water bath for 2 h. Afterward, the saponified mixtures were extracted with 0.6 mL n-heptane. n-heptane was evaporated by vacuum centrifugation. Dried samples were dissolved in 0.5 mL of acetonitrile for the analysis using Shimadzu HPLC system equipped with UV detector (Shimadzu SPD-20A) and C18 column (Phenomenex Kinetex 5 μL C18). The mobile phase was 100% acetonitrile at a flow rate of 2 mL/min. Ergosterol was detected at 280 nm. Data were collected by Shimadzu LabSolutions CS C191-E020.

TAL was quantified by HPLC (LC-20A, Shimadzu, Kyoto, Japan) equipped with a UV detector (Shimadzu SPD-20A, Shimadzu, Kyoto, Japan). Specifically, the fermentation broth was diluted at an appropriate rate, mixed thoroughly, and centrifuged at 21,130 × *g* to remove cells. The supernatant was injected into HPLC

and separated using a C18 column (Phenomenex Kinetex 5 μL C18, Phenomenex Inc, Torrance, CA) at 40 °C. TAL was detected at 280 nm. A dual mobile phase condition with 90 % of mobile phase A: 1% acetic acid in the water, and 10 % of mobile phase B: 1% acetic acid in acetonitrile, was adopted at a flow rate of 1 mL/min. A standard curve was prepared using >98.0% purity TAL from Sigma-Aldrich (Cat. No. H43415). Notably, for fed-batch bioreactor samples with TAL precipitation, the culture broth was serially diluted at increasing rates until the quantification results stabilized.

β-carotene and vitamin A from SR8A cultures were extracted and analyzed as follows[34]. First, cell pellets were crushed by a beat beater with acetone as an extractive solvent. Cell extracts, culture media, and dodecane layers of two-phase fermentation were diluted and analyzed using HPLC (LC-20A, Shimadzu, Kyoto, Japan) equipped with a UV detector (Shimadzu SPD-20A, Shimadzu, Kyoto, Japan). The mobile phase was 95% methanol and 5% acetonitrile at a flow rate of 1 ml/min. β-carotene and vitamin A were separated using a C18 column (Phenomenex Kinetex 5 μL C18, Phenomenex Inc, Torrance, CA) at 40 °C and detected at 453 nm and 352 nm, respectively. Data from the above quantitative analysis were evaluated for statistical significance of the differences by one-way ANOVA followed by Tukey's multiple-comparison test where necessary using Microsoft Excel 2016 and SigmaPlot 14.

**Reporting summary**. Further information on research design is available in the Nature Research Reporting Summary linked to this article.

## Data availability

Data supporting the findings of this work are available within the paper and its Supplementary Information files. A reporting summary for this article is available as a Supplementary Information file. RNA-seq data of *S. cerevisiae* SR7 are available under NCBI BioProject PRJNA748193 and BioSample accessions SAMN20309334 and SAMN20309335. Source data are provided with this paper.

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

## Acknowledgements

This work is supported by the DOE Center for Advanced Bioenergy and Bioproducts Innovation (U.S. Department of Energy, Office of Science, Office of Biological and Environmental Research under Award Number DE-SC0018420). Any opinions, findings, and conclusions, or recommendations expressed in this publication are those of the author(s) and do not necessarily reflect the views of the U.S. Department of Energy. The authors thank Dr. Guochang Zhang for his suggestions on experimental design and Christine Anne Atkinson for her diligent proofreading. L.S. would like to thank the China Scholarship Council for financial support (File No. 201606350094).

## Author contributions

L.S. and Y.S.J. designed research; L.S., J.W.L., S.Y., Z.Q.S., and S.R.K. performed research; L.S., S.R.K., and S.L. analyzed data; and L.S., S.L., and Y.S.J. wrote the paper. All authors read and approved the paper.

## Competing interests

The authors declare no competing interest.
