## [Peer Review File · Nature Communications]

REVIEWER COMMENTS

Reviewer #1 (Remarks to the Author):

The manuscript entitled "Complete and efficient conversion of plant cell wall hemicellulose into high-value bioproducts by engineered yeast" by Sun et al, reports the engineering of *S. cerevisiae* for xylose assimilation for the production of b-carotene and triacetic acid lactone.

Of the three aspects tackled in the manuscript 1) xylose consumption, 2) acetate consumption and 3) b-carotene and triacetic acid lactone, none is really new.

For over two decades yeast scientists have worked and engineered several pathways for xylose assimilation. There are plethora of examples of organic compounds already made out of xylose. This aspect have been covered is a recent review by the the authors of the submitted manuscript. Additionally both b-carotene and triacetic acid lactone have already been engineered not only in *S. cerevisiae*.

Finally acetic acid assimilation for detoxification of lignocellulosic hydrolysate has also been investigated, an aspect of literature not described in the introduction of the manuscript.

Next to this, this is a well conducted study, accurate, but not novel enough.

Reviewer #2 (Remarks to the Author):

In this paper, the authors showed that engineered xylose-assimilating *Saccharomyces cerevisiae* strain co-consumed xylose and acetate under aerobic conditions, thus improving synthesis of acetyl-CoA derived chemicals such as triacetic acid lactone and vitamin A. Xylose and acetate are the main carbon sources of hemicellulose hydrolysate. Acetate was previously considered a fermentation inhibitor, but this study proposes a new solution that utilizes this inhibitor as a useful carbon source providing cytoplasmic acetyl-CoA, a precursor of various value-added products. The following is some comments on this paper.

1. Fig. 2a: The SR8 strain used in this study has an inactive ALD6 gene. Because Ald6 is involved in ethanol utilization, the ALD6 mutation might affect ethanol utilization in the presence of acetate. Therefore, please clarify the effect of ALD6 mutation by investigating the co-utilization of glucose and acetate in cells with wild-type ALD6 such as SR7.
2. Fig. 3: The experiments shown in Fig. 2 were done with SR8 strain, but transcriptional studies (Fig. 3) were done with SR7. Is there any reason for this? In addition, the culture conditions and sampling time points for mRNA analysis should be indicated. Gene expression patterns may vary depending on the glucose concentration during growth.
3. P12, lines 241-244: The authors suggest that ethanol-dependent repression of ACS1 may be one of the reasons for the defect in acetate assimilation in the presence of glucose. However, because Acs1 is also necessary for ethanol assimilation, it is unlikely that ethanol inhibits its own degradation. In addition, such mechanisms may contradict to normal ethanol assimilation in cells grown only on glucose (for example, the first graph of Fig. 2a).
4. Fig. 4b: Please also show the titer of TAL.
4. P7, line 135. It would be helpful to explain the meaning of the 'inactivated over-flow metabolism'.
5. P8, lines 153, 154, 156: Please correct g/gDCW to mg/gDCW.

Reviewer #3 (Remarks to the Author):

The manuscript by Sun et al. establishes an innovative strategy to produce acetyl-CoA derived products in the yeast *Saccharomyces cerevisiae* by enabling the co-consumption of xylose, an hemicellulosic derived sugar, and acetate, a relevant microbial inhibitor. These are very interesting results in the quest for biobased products and for the development of a bioeconomy and thus should be of interest in a wider field. Overall, the article is very well-written, being clear and concise. The results achieved are significant regarding acetate metabolism. The conclusions attained in this work are strongly backed by extensive analysis of different process configurations, being well structured in a logical sequence. The proof of concept chosen, leading to the highest yield of TAL produced so far, clearly shows the advantages of co-utilization of xylose and acetic acid, an alternative process configuration. Therefore, I recommend this manuscript for publication. Some suggestions are given below for improvement:

- The introduction is very elucidative regarding the advantages of acetate metabolism in a context of valorization of lignocellulosic materials. But as oleaginous yeasts have been successfully used to produce high-value acetyl-CoA-derived molecules, as stated, why would it be advantageous to use *S. cerevisiae* instead? I believe that it would be beneficial to elaborate a little bit on that.
- Was xylitol production analysed during the experiments? Xylitol formation can be reduced by providing oxygen still I guess the authors have monitored xylitol. If no production was observed this should be clearly stated
- The results with switchgrass hemicellulose hydrolysate led to glycerol production. Have the authors use the hydrolysate as-is or less concentrated?
- Line 53: change "an NADH-dependent" to "a NADH-dependent"
- Line 54: remove the comma between "acetate" and "and improved ethanol"
- Lines 55-59: this sentence is long and has several "and" that does not aid to its readability. Maybe it could be split in two.
- Line 68: remove the comma between "transporter20" and "or"
- Line 70: remove the comma between "cycle" and "or"
- Line 73: remove the comma between "(ROS)23" and "and"
- Line 77: remove the comma between "transport" and "and"
- Line 134: remove at high concentrations
- Line 135: change "over-flow metabolism" to "overflow metabolism"
- Line 154/155: I would suggest removing the sentence "Ergosterol content is also substantially increased by xylose and acetate co-utilization" as it is already stated a few lines above (Line 150)
- Lines 206-210: I believe that the handling and characterization of the hemicellulosic hydrolysate would be more appropriately placed in the material and methods section.
- Line 294: change "over-flow metabolism" to "overflow metabolism"
- Line 305: change "principle" to "principal" (or equivalent)
- Line 308: change "a" to "an"
- Line 332: the first two sentences should be merged.
- Line 364: Define Lpm
- Statistical analysis methodology is lacking in the Materials and methods section
- Fig. 1 legend is lacking explanation for the different types of arrows: meaning of color, bold/not bold arrows, etc.
- Fig.1 I do not see the point of having glucose pointing to Food?
- It would be appropriate to do statistical analysis to specific data in Fig. 2, 4 and 7 as it was made for Fig. 3, to a more reliable comparison between the data
- Figures S4 and S5 on the SI appendix are missing the labelling for each compound represented in the plots.

Responses to Reviewer's comments

Reviewers' comments are copied in black while authors' responses are written in blue.

We are grateful to the reviewers for performing a peer review of our manuscript. The kind and insightful comments by the reviewers enabled us to improve the manuscript significantly. We have conducted necessary experiments and revised the manuscript based on the reviewers' comments. Changes are highlighted in blue in the Main Text and Supplementary Information files.

Reviewer #1 (Remarks to the Author):

The manuscript entitled "Complete and efficient conversion of plant cell wall hemicellulose into high-value bioproducts by engineered yeast" by Sun et al, reports the engineering of *S. cerevisiae* for xylose assimilation for the production of b-carotene and triacetic acid lactone.

Of the three aspects tackled in the manuscript 1) xylose consumption, 2) acetate consumption and 3) b-carotene and triacetic acid lactone, none is really new.

For over two decades yeast scientists have worked and engineered several pathways for xylose assimilation. There are plethora of examples of organic compounds already made out of xylose. This aspect have been covered is a recent review by the the authors of the submitted manuscript. Additionally both b-carotene and triacetic acid lactone have already been engineered not only in *S. cerevisiae*.

Finally acetic acid assimilation for detoxification of lignocellulosic hydrolysate has also been investigated, an aspect of literature not described in the introduction of the manuscript.

Next to this, this is a well conducted study, accurate, but not novel enough.

Response: Thank you for your comments. In this study, we proposed a strategy that effectively transforms a fermentation inhibitor acetate which is prevalent in lignocellulosic hydrolysates into a favorable substrate for the enhanced supply of cytosolic acetyl-CoA in engineered yeast, enabling high-level production of various acetyl-CoA derivatives and complete bioconversion of hemicellulose fractions. We hope the reviewer to find the novelty of this study considering the following aspects:

1. Previous efforts for detoxifying acetate in lignocellulosic hydrolysates have been devoted to improving acetate tolerance of yeast, relieving but being unable to eliminate its inhibitory and toxic effects on cellular physiology (Chen et al., 2016; Cunha et al., 2018, p. 1; Ko et al., 2020; Mira et al., 2010; Oh et al., 2019). According to our extensive literature search, nearly all previous studies reporting acetate assimilation by *S. cerevisiae* in lignocellulosic hydrolysates implemented a reductive acetate assimilation pathway which is operational only under strict anaerobic conditions. As such, the acetate reduction pathway has limited capacity and is only applicable to bioethanol production as it converts acetate into ethanol (Guadalupe Medina et al., 2010; Henningsen et al., 2015; Papapetridis et al., 2016, 2017; Wei et al., 2013,

2015; Zhang et al., 2016). This aspect was covered in the **Introduction (Line 55-61)**. For the first time, we discovered that acetate at high levels (~ 12.38 g/L) can be co-consumed with xylose by engineered *S. cerevisiae* under aerobic conditions without any inhibitory effects on cell growth and xylose utilization (**Fig. 2**), providing a new and more efficient solution to detoxify and utilize acetate in lignocellulose hydrolysate as a carbon source.

2. It is true that xylose assimilation by engineered *S. cerevisiae* has been extensively studied over the past two decades, yielding a number of efficient xylose-fermenting strains (Kwak & Jin, 2017) and a deep understanding of the unique traits of yeast xylose metabolism (Kwak et al., 2019). However, there was no report delving into the regulation of acetate consumption during xylose metabolism while xylose and acetate are inevitably co-present in hemicellulose hydrolysates, even though glucose repression on acetate uptake has been known for over twenty years (Casal et al., 1996). By investigating the mechanisms enabling rapid xylose and acetate co-utilization, we demonstrated that xylose metabolism does not pose any repression on acetate utilization due to its inactive overflow metabolism and active respiratory pathways under aerobic conditions (**Fig. 2** and **3**). Instead, xylose metabolism promotes acetate co-consumption by supporting cell growth and supplying ATP (**Fig. 1**). These results provide new understanding of yeast xylose metabolism and open doors for yeast substrate extension as well as a bioconversion paradigm of sugar and organic acid co-utilization.
3. A key limitation for the biosynthesis of high-value products, such as specialty lipids, sterols, isoprenoids, and polyketides, in *S. cerevisiae* is the poor availability of a precursor cytosolic acetyl-CoA (Chen et al., 2013). In this study, we demonstrated strong synergies between acetate and xylose metabolism in engineered yeast which leads to not only better cell growth and xylose consumption but also a unique metabolic re-configuration that enhances cytosolic acetyl-CoA supply (**Fig. 2c**). Based on this metabolic re-configuration, we established a metabolic engineering strategy to produce acetyl-CoA derivatives using xylose and acetate as dual-substrates. Triacetic acid lactone (TAL), a polyketide, and vitamin A, an isoprenoid, which have been produced in different microbial hosts (Jang et al., 2011; Li et al., 2018; Markham et al., 2018; Saunders et al., 2015; Sun et al., 2019) were chosen as testbed molecules to examine the effectiveness and versatility of the dual-substrate strategy. The highest biosynthetic productivity of TAL and significantly improved titers of vitamin A obtained here suggest that the dual-substrate strategy is highly effective and promising for the production of acetyl-CoA derived molecules from hemicellulose hydrolysates. Also, to the best of our knowledge, this is the first report to synthesize TAL from xylose or in a lignocellulosic hydrolysate by *S. cerevisiae*.
4. Our strategy is particularly meaningful in that it can be seamlessly integrated into cellulosic biomass-based biorefineries as acetate and xylose are predominant components of hemicellulose hydrolysates, which can be readily obtained from lignocellulose materials via thermochemical processing without the need of costly enzymatic hydrolysis (Hassan et al., 2018). Using TAL production as an example, we demonstrated that substrates in hemicellulose hydrolysates can be completely converted into acetyl-CoA derived fine chemicals (**Fig. 6**). Given the simplicity of pretreatment process needed, the effectiveness of

bioconversion, and the high value of final products, our strategy would enable economic bioconversion of cellulosic biomass.

Overall, we believe that this study reports significant findings and will be interesting to a broad audience. We noted that the aspect of engineering acetate tolerance in yeast was missing in the **Introduction**. Thus, we added the information in **Line 53-55** and updated the **References**.

Line 53-55, “In addition to these pre-fermentation strategies, efforts have been devoted to engineering acetate tolerance in yeast, alleviating but being unable to eliminate the inhibitory and toxic effects of acetate^{13,14}.”

Reviewer #2 (Remarks to the Author):

In this paper, the authors showed that engineered xylose-assimilating *Saccharomyces cerevisiae* strain co-consumed xylose and acetate under aerobic conditions, thus improving synthesis of acetyl-CoA derived chemicals such as triacetic acid lactone and vitamin A. Xylose and acetate are the main carbon sources of hemicellulose hydrolysate. Acetate was previously considered a fermentation inhibitor, but this study proposes a new solution that utilizes this inhibitor as a useful carbon source providing cytoplasmic acetyl-CoA, a precursor of various value-added products. The following is some comments on this paper.

1. Fig. 2a: The SR8 strain used in this study has an inactive ALD6 gene. Because Ald6 is involved in ethanol utilization, the ALD6 mutation might affect ethanol utilization in the presence of acetate. Therefore, please clarify the effect of ALD6 mutation by investigating the co-utilization of glucose and acetate in cells with wild-type ALD6 such as SR7.

Response: Thank you for the insightful comment. According to the suggestion, we conducted additional experiment to investigate the effect of *ALD6* deletion on ethanol reassimilation in the presence of acetate by culturing SR7 strain which has wild-type *ALD6* with glucose and acetate at various concentrations. All the media and culturing conditions were kept the same as the SR8 culturing experiments.

We included the culture profiles of the SR7 strain in the **Supplementary Figure 1** and below for easier comparison with those of the SR8 strain. In concordance with your assumption, the SR7 strain was able to respire ethanol much faster than the SR8 strain in both the presence and absence of acetate, suggesting that the disrupted *ALD6* did negatively affect ethanol utilization by the SR8 strain. Thus, we stated the effect of *ALD6* mutation on ethanol reassimilation in the **Results** section (**Line 115-118**).

Line 115-118, “The inhibited ethanol reassimilation was partially attributed to the disruption of *ALD6* gene involved in ethanol utilization as the parental strain SR7 with wildtype *ALD6* respired ethanol much faster than the SR8 strain (**Supplementary Figure 1**).”

Glucose cultures of **SR8** and **SR7** strain with acetate supplementation of 0, 5, 10, and 15 g/L.

However, the intact *ALD6* gene in SR7 did not help with acetate consumption that was strongly inhibited by glucose metabolism.

2. Fig. 3: The experiments shown in Fig. 2 were done with SR8 strain, but transcriptional studies (Fig. 3) were done with SR7. Is there any reason for this? In addition, the culture conditions and sampling time points for mRNA analysis should be indicated. Gene expression patterns may vary depending on the glucose concentration during growth.

Response: The SR8 strain had been constructed previously through evolutionary engineering of the SR7 strain and subsequent disruption of *ALD6* so that it demonstrated improved xylose metabolism compared with the parental strain SR7 (Kim et al., 2013). Besides the *ALD6* knockout, the SR8 strain accumulated around 20 other mutations during the adaptive laboratory evolution on xylose. As the effects of those mutations in SR8 on xylose metabolism are not fully understood, we conducted transcriptional studies based on the parental strain SR7 which has a clear genetic background in order to rule out the potential effects of those mutations and reveal the general mechanisms enabling xylose and acetate co-consumption by engineered yeast (**Line 396-399**). However, we believe the uninvestigated mutations in SR8 are not necessary for the co-utilization of xylose and acetate because the rationally engineered Tal4 strain with a clear genotype was able to co-consume xylose and acetate efficiently (**Supplementary Figure 6**).

As suggested by the reviewer, we added information regarding the culture conditions and sampling time for RNA sequencing in the **Materials and Methods** section (**Line 399-403**) and included the culturing profiles in the **Supplementary Figure 9**. Cells were sampled for mRNA analysis when the absorbance at 600 nm (OD_{600}) reached 1. By the time of sampling, there was around 37 g/L of glucose or 33 g/L of xylose remained in the media.

3. P12, lines 241-244: The authors suggest that ethanol-dependent repression of ACS1 may be one of the reasons for the defect in acetate assimilation in the presence of glucose. However, because *Acs1* is also necessary for ethanol assimilation, it is unlikely that ethanol inhibits its own

degradation. In addition, such mechanisms may contradict to normal ethanol assimilation in cells grown only on glucose (for example, the first graph of Fig. 2a).

Response: We carefully reviewed this assumption by reading related papers. In *Saccharomyces cerevisiae*, ethanol assimilation is initiated through the ‘pyruvate-dehydrogenase bypass’ involving the conversion of acetate to acetyl-CoA catalyzed by two acetyl-CoA synthetases (*ACSI* and *ACS2*) (Kozak et al., 2016). It is reasonable that ethanol would not restrain its own catabolism by repressing both *ACSI* and *ACS2* genes. However, transcription of *ACSI* is subject to ethanol repression according to previous reports (Berg et al., 1996; Kratzer & Schüller, 1995). Besides, $\Delta acs1$ mutants constructed in various strain backgrounds did not exhibit growth defects on media containing ethanol as a sole carbon source (De Virgilio et al., 1992; Kratzer & Schüller, 1995), which explains the normal ethanol re-assimilation by the cells grown only on glucose. Despite the normal utilization of ethanol as the sole carbon source, we assume that ethanol-dependent repression of *ACSI* might exert negative effects on acetate consumption when ethanol and acetate co-exist and compete with each other for ACS activities. This assumption is in line with the results in **Fig. 2a** and **Supplementary Figures 1&2** regarding the hindered acetate and ethanol assimilation.

4. Fig. 4b: Please also show the titer of TAL.

Response: We added the titer of TAL in **Fig. 4b** as suggested by the reviewer.

4. P7, line 135. It would be helpful to explain the meaning of the ‘inactivated over-flow metabolism’.

Response: We explained the meaning of the ‘inactivated overflow metabolism’ in the **Results** section (**Line 143-144**).

5. P8, lines 153, 154, 156: Please correct g/gDCW to mg/gDCW.

Response: Thank you. We made the corrections (**Line 161, 162, 163**).

Reviewer #3 (Remarks to the Author):

The manuscript by Sun et al. establishes an innovative strategy to produce acetyl-CoA derived products in the yeast *Saccharomyces cerevisiae* by enabling the co-consumption of xylose, an hemicellulosic derived sugar, and acetate, a relevant microbial inhibitor. These are very interesting results in the quest for biobased products and for the development of a bioeconomy and thus should be of interest in a wider field. Overall, the article is very well-written, being clear and concise. The results achieved are significant regarding acetate metabolism. The conclusions attained in this work are strongly backed by extensive analysis of different process configurations, being well structured in a logical sequence. The proof of concept chosen, leading to the highest yield of TAL produced so far, clearly shows the advantages of co-utilization of xylose and acetic acid, an alternative process configuration. Therefore, I recommend this manuscript for publication. Some suggestions are given below for improvement:

Response: We sincerely appreciate the reviewer for the positive and encouraging comments.

- The introduction is very elucidative regarding the advantages of acetate metabolization in a context of valorization of lignocellulosic materials. But as oleaginous yeasts have been

successfully used to produce high-value acetyl-CoA-derived molecules, as stated, why would it be advantageous to use *S. cerevisiae* instead? I believe that it would be beneficial to elaborate a little bit on that.

Response: Thank you for the suggestion. We underscored the importance of acetate detoxification and valorization by *S. cerevisiae* instead of nonconventional yeasts in the **Introduction (Line 62-64)**.

Line 62-64, “Enabling efficient acetate detoxification and valorization by *S. cerevisiae* would greatly facilitate industrial cellulosic biorefineries due to the versatility and robustness of *S. cerevisiae* as an industrial production platform.”

- Was xylitol production analysed during the experiments? Xylitol formation can be reduced by providing oxygen still I guess the authors have monitored xylitol. If no production was observed this should be clearly stated.

Response: Yes, we monitored xylitol production during the experiments but did not observe xylitol accumulation in the aerobic cultures. We included this information in the **Results** section (**Line 222-224**)

Line 222-224, “We did not observe xylitol accumulation in this and all other xylose culturing experiments, suggesting sufficient oxygen supply to eliminate xylitol production.”

- The results with switchgrass hemicellulose hydrolysate led to glycerol production. Have the authors use the hydrolysate as-is or less concentrated?

Response: We have also cultured the Tal4 strain in unconcentrated switchgrass hemicellulose hydrolysate with an initial cell density of $OD_{600nm} = 1$, obtaining a TAL titer of 2377.60 mg/L. As shown below, the cells consumed sugars and acetate much slower than when cultured in 2 times concentrated hydrolysate with an initial cell density of $OD_{600nm} = 10$ (**Fig. 6**) but did not accumulate glycerol at the end of cultures, indicating less osmotic stress on the cells. This also led to a higher yield of TAL in 1x hydrolysate (86.53 mg/g substrates) than that (67.45 mg/g substrates) in 2x hydrolysate.

Tal4 in 1x hydrolysate at $OD_{600nm} 1$

Tal4 in 2x hydrolysate at $OD_{600nm} 10$

Production of TAL by Tal4 strain in unconcentrated or 2 times concentrated switchgrass hemicellulose hydrolysate. Profiles of TAL, xylose, acetate, glucose, cell biomass, ethanol, and glycerol concentrations are presented.

We included the culturing profiles of Tal4 strain in unconcentrated switchgrass hemicellulose hydrolysate with initial cell density of OD_{600nm} 1 in the **Supplementary Information** as **Supplementary Figure 7**. The corresponding description was added to the **Results** section (**Line 219-222**).

Line 219-222, “When cultured in unconcentrated hydrolysate, the strain did not accumulate glycerol at the end of fermentation, leading to a higher yield of TAL (86.53 mg/g substrates) (**Supplementary Figure 7**).”

- Line 53: change “an NADH-dependent” to “a NADH-dependent”

Response: We made the correction (**Line 55**).

- Line 54: remove the comma between “acetate” and “and improved ethanol”

Response: We removed the comma as suggested (**Line 57**).

- Lines 55-59: this sentence is long and has several “and” that does not aid to its readability. Maybe it could be split in two.

Response: Thank you for the suggestion. We split the sentence into two (**Line 58-61**).

- Line 68: remove the comma between “transporter20” and “or”

Response: We removed the comma as suggested (**Line 73**).

- Line 70: remove the comma between “cycle” and “or”

Response: We removed the comma as suggested (**Line 75**).

- Line 73: remove the comma between “(ROS)23” and “and”

Response: We removed the comma as suggested (**Line 78**).

- Line 77: remove the comma between “transport” and “and”

Response: We removed the comma as suggested (**Line 81**).

- Line 134: remove at high concentrations

Response: For a better clarity, we changed “at high concentrations (> 4.25 g/L)” to “at a concentration higher than 4.25 g/L” (**Line 141**).

- Line 135: change “over-flow metabolism” to “overflow metabolism”

Response: Thank you. We made the correction (**Line 143**).

- Line 154/155: I would suggest removing the sentence “Ergosterol content is also substantially increased by xylose and acetate co-utilization” as it is already stated a few lines above (Line 150)

Response: As suggested by the reviewer, we removed the redundant sentence (**Line 162**).

- Lines 206-210: I believe that the handling and characterization of the hemicellulosic hydrolysate would be more appropriately placed in the material and methods section.

Response: As suggested by the reviewer, we moved the information regarding the handling and characterization of the hemicellulose hydrolysate in the Results section to the Materials and Methods section (**Line 387-394**).

- Line 294: change “over-flow metabolism” to “overflow metabolism”

Response: We made the correction (**Line 306**).

- Line 305: change “principle” to “principal” (or equivalent)

Response: We changed “principle” to “principal” as suggested (**Line 317**).

- Line 308: change “a” to “an”

Response: We think “a ubiquitous” is correct instead of “an ubiquitous” (**Line 320**).

- Line 332: the first two sentences should be merged.

Response: We merged the two sentences as suggested (**Line 344**).

- Line 364: Define Lpm

Response: We define “Lpm” as “Lpm (lines per minute)” (**Line 377**).

- Statistical analysis methodology is lacking in the Materials and methods section

Response: We added the statistical analysis methodology in the **Materials and Methods** section (**Line 411-412; Line 461-463**).

Line 411-412, “Statistical significance of the differences was evaluated using two-tailed Student’s t-tests.”

Line 461-463, “Data from above quantitative analysis were evaluated for statistical significance of the differences by one-way ANOVA followed by Tukey's multiple-comparison test where necessary.”

- Fig. 1 legend is lacking explanation for the different types of arrows: meaning of color, bold/not bold arrows, etc.

Response: We added explanation for the different types of arrows in the legend for Fig. 1 (**Line 628-631**).

Line 628-631, “Color code for arrows: grey, glucose metabolism; green, xylose metabolism; orange, acetate metabolism; black, other endogenous pathways; black in bold, reaction with strong flux; blue, heterologous TAL or vitamin A biosynthetic pathway.”

- Fig.1 I do not see the point of having glucose pointing to Food?

Response: We removed “Food” and the arrow pointing to it in **Fig. 1**.

- It would be appropriate to do statistical analysis to specific data in Fig. 2, 4 and 7 as it was made for Fig. 3, to a more reliable comparison between the data

Response: We conducted one-way ANOVA analysis followed by Tukey's multiple-comparison test for specific data in **Fig. 2, 4 and 7**. The statistical significance was marked in the figures using “*(p < 0.05)”, “**(p < 0.01)”, and “*** (p < 0.005)”.

- Figures S4 and S5 on the SI appendix are missing the labelling for each compound represented in the plots.

Response: We added the labelling for each compound represented in the plots in **Supplementary Figure 5 and 6**. Thank you!

References cited

- Berg, M. A. van den, Jong-Gubbels, P. de, Kortland, C. J., Dijken, J. P. van, Pronk, J. T., & Steensma, H. Y. (1996). The Two Acetyl-coenzyme A Synthetases of *Saccharomyces cerevisiae* Differ with Respect to Kinetic Properties and Transcriptional Regulation. *Journal of Biological Chemistry*, 271(46), 28953–28959. <https://doi.org/10.1074/jbc.271.46.28953>
- Casal, M., Cardoso, H., & Leao, C. (1996). Mechanisms regulating the transport of acetic acid in *Saccharomyces cerevisiae*. *Microbiology*, 142(6), 1385–1390. <https://doi.org/10.1099/13500872-142-6-1385>
- Chen, Y., Daviet, L., Schalk, M., Siewers, V., & Nielsen, J. (2013). Establishing a platform cell factory through engineering of yeast acetyl-CoA metabolism. *Metabolic Engineering*, 15, 48–54. <https://doi.org/10.1016/j.ymben.2012.11.002>
- Chen, Y., Stabryla, L., & Wei, N. (2016). Improved Acetic Acid Resistance in *Saccharomyces cerevisiae* by Overexpression of the WHI2 Gene Identified through Inverse Metabolic Engineering. *Applied and Environmental Microbiology*, 82(7), 2156–2166. <https://doi.org/10.1128/AEM.03718-15>
- Cunha, J. T., Costa, C. E., Ferraz, L., Romani, A., Johansson, B., Sá-Correia, I., & Domingues, L. (2018). HAA1 and PRS3 overexpression boosts yeast tolerance towards acetic acid improving xylose or glucose consumption: Unravelling the underlying mechanisms. *Applied Microbiology and Biotechnology*, 102(10), 4589–4600. <https://doi.org/10.1007/s00253-018-8955-z>
- De Virgilio, C., Bürckert, N., Barth, G., Neuhaus, J. M., Boller, T., & Wiemken, A. (1992). Cloning and disruption of a gene required for growth on acetate but not on ethanol: The acetyl-coenzyme A synthetase gene of *Saccharomyces cerevisiae*. *Yeast (Chichester, England)*, 8(12), 1043–1051. <https://doi.org/10.1002/yea.320081207>

- Guadalupe Medina, V., Almering, M. J. H., van Maris, A. J. A., & Pronk, J. T. (2010). Elimination of glycerol production in anaerobic cultures of a *Saccharomyces cerevisiae* strain engineered to use acetic acid as an electron acceptor. *Applied and Environmental Microbiology*, *76*(1), 190–195. <https://doi.org/10.1128/AEM.01772-09>
- Hassan, S. S., Williams, G. A., & Jaiswal, A. K. (2018). Emerging technologies for the pretreatment of lignocellulosic biomass. *Bioresource Technology*, *262*, 310–318. <https://doi.org/10.1016/j.biortech.2018.04.099>
- Henningsen, B. M., Hon, S., Covalla, S. F., Sonu, C., Argyros, D. A., Barrett, T. F., Wiswall, E., Froehlich, A. C., & Zelle, R. M. (2015). Increasing anaerobic acetate consumption and ethanol yields in *Saccharomyces cerevisiae* with NADPH-specific alcohol dehydrogenase. *Applied and Environmental Microbiology*, *81*(23), 8108–8117. <https://doi.org/10.1128/AEM.01689-15>
- Jang, H.-J., Yoon, S.-H., Ryu, H.-K., Kim, J.-H., Wang, C.-L., Kim, J.-Y., Oh, D.-K., & Kim, S.-W. (2011). Retinoid production using metabolically engineered *Escherichia coli* with a two-phase culture system. *Microbial Cell Factories*, *10*(1), 59. <https://doi.org/10.1186/1475-2859-10-59>
- Kim, S. R., Skerker, J. M., Kang, W., Lesmana, A., Wei, N., Arkin, A. P., & Jin, Y.-S. (2013). Rational and Evolutionary Engineering Approaches Uncover a Small Set of Genetic Changes Efficient for Rapid Xylose Fermentation in *Saccharomyces cerevisiae*. *PLOS ONE*, *8*(2), e57048. <https://doi.org/10.1371/journal.pone.0057048>
- Ko, J. K., Enkh-Amgalan, T., Gong, G., Um, Y., & Lee, S.-M. (2020). Improved bioconversion of lignocellulosic biomass by *Saccharomyces cerevisiae* engineered for tolerance to acetic acid. *GCB Bioenergy*, *12*(1), 90–100. <https://doi.org/10.1111/gcbb.12656>
- Kozak, B. U., van Rossum, H. M., Niemeijer, M. S., van Dijk, M., Benjamin, K., Wu, L., Daran, J.-M. G., Pronk, J. T., & van Maris, A. J. A. (2016). Replacement of the initial steps of ethanol metabolism in *Saccharomyces cerevisiae* by ATP-independent acetylating acetaldehyde dehydrogenase. *FEMS Yeast Research*, *16*(2), fow006. <https://doi.org/10.1093/femsyr/fow006>
- Kratzer, S., & Schüller, H.-J. (1995). Carbon source-dependent regulation of the acetyl-coenzyme A synthetase-encoding gene ACSI from *Saccharomyces cerevisiae*. *Gene*, *161*(1), 75–79. [https://doi.org/10.1016/0378-1119\(95\)00289-1](https://doi.org/10.1016/0378-1119(95)00289-1)
- Kwak, S., & Jin, Y.-S. (2017). Production of fuels and chemicals from xylose by engineered *Saccharomyces cerevisiae*: A review and perspective. *Microbial Cell Factories*, *16*(1), 82. <https://doi.org/10.1186/s12934-017-0694-9>
- Kwak, S., Jo, J. H., Yun, E. J., Jin, Y.-S., & Seo, J.-H. (2019). Production of biofuels and chemicals from xylose using native and engineered yeast strains. *Biotechnology Advances*, *37*(2), 271–283. <https://doi.org/10.1016/j.biotechadv.2018.12.003>
- Li, Y., Qian, S., Dunn, R., & Cirino, P. C. (2018). Engineering *Escherichia coli* to increase triacetic acid lactone (TAL) production using an optimized TAL sensor-reporter system. *Journal of Industrial Microbiology & Biotechnology*, *45*(9), 789–793. <https://doi.org/10.1007/s10295-018-2062-0>
- Markham, K. A., Palmer, C. M., Chwatko, M., Wagner, J. M., Murray, C., Vazquez, S., Swaminathan, A., Chakravarty, I., Lynd, N. A., & Alper, H. S. (2018). Rewiring *Yarrowia lipolytica* toward triacetic acid lactone for materials generation. *Proceedings of the National Academy of Sciences*, *115*(9), 2096–2101. <https://doi.org/10.1073/pnas.1721203115>

- Mira, N. P., Palma, M., Guerreiro, J. F., & Sá-Correia, I. (2010). Genome-wide identification of *Saccharomyces cerevisiae* genes required for tolerance to acetic acid. *Microbial Cell Factories*, *9*, 79. <https://doi.org/10.1186/1475-2859-9-79>
- Oh, E. J., Wei, N., Kwak, S., Kim, H., & Jin, Y.-S. (2019). Overexpression of RCK1 improves acetic acid tolerance in *Saccharomyces cerevisiae*. *Journal of Biotechnology*, *292*, 1–4. <https://doi.org/10.1016/j.jbiotec.2018.12.013>
- Papapetridis, I., van Dijk, M., Dobbe, A. P. A., Metz, B., Pronk, J. T., & van Maris, A. J. A. (2016). Improving ethanol yield in acetate-reducing *Saccharomyces cerevisiae* by cofactor engineering of 6-phosphogluconate dehydrogenase and deletion of ALD6. *Microbial Cell Factories*, *15*, 67. <https://doi.org/10.1186/s12934-016-0465-z>
- Papapetridis, I., van Dijk, M., van Maris, A. J. A., & Pronk, J. T. (2017). Metabolic engineering strategies for optimizing acetate reduction, ethanol yield and osmotolerance in *Saccharomyces cerevisiae*. *Biotechnology for Biofuels*, *10*(1), 107. <https://doi.org/10.1186/s13068-017-0791-3>
- Saunders, L. P., Bowman, M. J., Mertens, J. A., Silva, N. A. D., & Hector, R. E. (2015). Triacetic acid lactone production in industrial *Saccharomyces* yeast strains. *Journal of Industrial Microbiology & Biotechnology*, *42*. <https://doi.org/10.1007/s10295-015-1596-7>
- Sun, L., Kwak, S., & Jin, Y.-S. (2019). Vitamin A Production by Engineered *Saccharomyces cerevisiae* from Xylose via Two-Phase in Situ Extraction. *ACS Synthetic Biology*, *8*(9), 2131–2140. <https://doi.org/10.1021/acssynbio.9b00217>
- Wei, N., Oh, E. J., Million, G., Cate, J. H. D., & Jin, Y.-S. (2015). Simultaneous Utilization of Cellobiose, Xylose, and Acetic Acid from Lignocellulosic Biomass for Biofuel Production by an Engineered Yeast Platform. *ACS Synthetic Biology*, *4*(6), 707–713. <https://doi.org/10.1021/sb500364q>
- Wei, N., Quarterman, J., Kim, S. R., Cate, J. H. D., & Jin, Y.-S. (2013). Enhanced biofuel production through coupled acetic acid and xylose consumption by engineered yeast. *Nature Communications*, *4*, 2580. <https://doi.org/10.1038/ncomms3580>
- Zhang, G.-C., Kong, I. I., Wei, N., Peng, D., Turner, T. L., Sung, B. H., Sohn, J.-H., & Jin, Y.-S. (2016). Optimization of an acetate reduction pathway for producing cellulosic ethanol by engineered yeast. *Biotechnology and Bioengineering*, *113*(12), 2587–2596. <https://doi.org/10.1002/bit.26021>

REVIEWERS' COMMENTS

Reviewer #2 (Remarks to the Author):

All reviewers' comments are well addressed in the revised manuscript.

Reviewer #3 (Remarks to the Author):

I appreciate the authors' efforts to answer to the raised comments and consider that the manuscript content was improved. I do not have further comments

Responses to Reviewer's comments

Reviewers' comments are copied in black while authors' responses are written in blue.

Reviewer #2 (Remarks to the Author):

All reviewers' comments are well addressed in the revised manuscript.

Response: Thank you.

Reviewer #3 (Remarks to the Author):

I appreciate the authors' efforts to answer to the raised comments and consider that the manuscript content was improved. I do not have further comments

Response: Thank you.